# Three-Month vs. One-Year Detraining Effects after Multicomponent Exercise Program in Hypertensive Older Women

**DOI:** 10.3390/ijerph19052871

**Published:** 2022-03-01

**Authors:** Luis Leitão, Moacir Marocolo, Hiago L. R. de Souza, Rhai André Arriel, Yuri Campos, Mauro Mazini, Ricardo Pace Junior, Teresa Figueiredo, Hugo Louro, Ana Pereira

**Affiliations:** 1Sciences and Technology Department, Superior School of Education of Polytechnic Institute of Setubal, 2910-761 Setúbal, Portugal; teresa.figueiredo@ese.ips.pt (T.F.); ana.fatima.pereira@ese.ips.pt (A.P.); 2Life Quality Research Centre, 2040-413 Rio Maior, Portugal; 3Department of Physiology, Institute of Biological Sciences, Federal University of Juiz de Fora, Juiz de Fora 36036-900, Brazil; isamjf@gmail.com (M.M.); hlrsouza@gmail.com (H.L.R.d.S.); rhaiarriel@bol.com.br (R.A.A.); 4Post Graduate Program in Physical Education, Federal University of Juiz de Fora, Juiz de Fora 36036-900, Brazil; reiclauy@hotmail.com; 5Studies Research Group in Neuromuscular Responses, Federal University of Lavras, Lavras 37200-900, Brazil; 6Graduate Program in Physical Education—Sudamerica Faculty, Cataguases 36774-552, Brazil; personalmau@hotmail.com; 7Graduate Program of Physical Education of Fasar—Santa Rita Faculty, Conselheiro Lafaiete 36400-000, Brazil; ricardo.pace.jr@gmail.com; 8Sports Science School of Rio Maior, Polytechnic Institute of Santarém, 2040-413 Rio Maior, Portugal; hlouro@esdrm.ipsantarem.pt; 9Center in Sports Sciences, Health Sciences and Human Development (CIDESD), 5000-801 Vila Real, Portugal

**Keywords:** older adults, hypertension, dyslipidemia, detraining, multicomponent exercise

## Abstract

Background: Chronic diseases are the leading causes of death and disability in older women. Physical exercise training programs promote beneficial effects for health and quality of life. However, exercise interruption periods may be detrimental for the hemodynamic and lipidic profiles of hypertensive older women with dyslipidemia. Methods: Nineteen hypertensive older women with dyslipidemia (exercise group: 67.5 ± 5.4 years, 1.53 ± 3.42 m, 71.84 ± 7.45 kg) performed a supervised multicomponent exercise training program (METP) during nine months, followed by a one-year detraining period (DT), while fourteen hypertensive older women (control group: 66.4 ± 5.2 years, 1.56 ± 3.10 m, 69.38 ± 5.24 kg) with dyslipidemia kept their continued daily routine without exercise. For both groups, hemodynamic and lipidic profiles and functional capacities (FCs) were assessed four times: before and after the METP and after 3 and 12 months of DT (no exercise was carried out). Results: The METP improved hemodynamic and lipidic profiles (*p* < 0.05), while three months of DT decreased all (*p* < 0.05) parameters, with the exception of diastolic blood pressure (DBP). One year of DT significantly (*p* < 0.01) decreased systolic blood pressure (7.85%), DBP (2.29%), resting heart rate (7.95%), blood glucose (19.14%), total cholesterol (10.27%), triglycerides (6.92%) and FC—agility (4.24%), lower- (−12.75%) and upper-body strength (−12.17%), cardiorespiratory capacity (−4.81%) and lower- (−16.16%) and upper-body flexibility (−11.11%). Conclusion: Nine months of the exercise program significantly improved the hemodynamic and lipid profiles as well as the functional capacities of hypertensive older women with dyslipidemia. Although a detraining period is detrimental to these benefits, it seems that the first three months are more prominent in these alterations.

## 1. Introduction

Sedentary and rarely active lifestyles promote negative adaptations in the health of elderly women, contributing to the aggregation of declines caused by the aging process, and are considered one of the most significant risk factors for global mortality [1,2,3]. Physical inactivity, together with cardiovascular diseases, is the biggest cause of death in elderly women, and those whose risk factors are hypertension and dyslipidemia have factors that are necessary to take into account for a better quality of life in these age groups [4,5,6].

Knowing these deleterious effects, organizations worldwide, such as the American College of Sports Science, American Heart Association and National Strength and Condition Association, promote physical exercise as a tool to mitigate or reverse some of these negative effects. Their recommendations for the type of exercise in this population may include strength training, interval training, aerobic training and combined training performed at moderate to vigorous intensities [1,7,8,9,10]. Garcia-Hermoso et al. [11] concluded that long-term exercise interventions in older adults result in a reduced mortality risk in clinical populations and do not increase the overall risk for dropouts due to health problems. For individuals with hypertension, the literature shows positive effects from aerobic exercise for decreasing resting and ambulatory blood pressure (BP) and from a moderate intensity due to the practicality for most inactive adults with pre- to established hypertension [12,13]. According to Mariano et al., [14], chronic exercise not only reduces systolic blood pressure (SBP) and diastolic blood pressure (DBP) in rest but also helps hypertensive individuals control BP levels under stress and reduce peaks of hypertension.

A type of physical exercise that is widely practiced in this population is the multicomponent exercise training program (METP), which results from a combination of aerobic, strength and flexibility training and which promotes benefits to the health of the elderly [15,16,17,18]. In this age group, there are several barriers that influence exercise practices, but this exercise method results in high adhesion and adherence rates; one of the reasons for this is the socialization that results from group training in every session [19,20].

The benefits that this exercise promotes occur in elderly women both with and without chronic illnesses, but the vast majority of these community programs have interruptions that promote sedentary activity beyond exercise, so some of the participants do not return after these interruptions [21,22]. There are studies that show the negative effect of detraining (DT) after these programs, mostly the effect of 6 weeks to three months of detraining, but as to what happens with durations longer than three months, little information is known [23,24,25].

Thus, we intend to analyze the effect of detraining on active hypertensive older women over 12 months of interruption in the practice of physical exercise in terms of functional capacity (FC) and hemodynamic and lipid profiles.

## 2. Materials and Methods

### 2.1. Experimental Design

This was a twenty-one-month study consisting of a 9-month of exercise program and 12 months of detraining. To analyze the changes in FC and in the lipid and hemodynamic profiles of active hypertensive older women during 12 months of detraining, assessments were performed at baseline (before exercise), 9 months post-training, 3 months post-detraining and 12 months post-detraining. All measurements were conducted in the same location and by one experienced researcher.

### 2.2. Sample and Ethical Procedures

The participant inclusion criteria were values above normal blood pressure levels (study thresholds: SBP: ≥ 130 mm Hg and DBP: ≥ 80 mm Hg), the ability to practice exercise without contraindication and attendance of at least 75% of the sessions. Participants with any osteoarticular problems that could restrict the execution of exercises, heart problems where the practice of exercise could injure their health, medical contraindications (e.g., surgeries) and who were already involved in any physical activity program were excluded. All procedures of the study were executed in accordance with the Declaration of Helsinki and approved for human experiments by local institutional ethical committee. Previous to study’s experimental protocols, the participants each visited their family doctor for medical evaluations and signed an informed consent form. From thirty-nine functionally independent older women who volunteered to participate, thirty-two completed this nineteen-month trial (seven dropped out of the study due to low adherence to the sessions), were instructed to keep, over the study, their normal lifestyle (dietary and physical activity routines) and were prohibited during the last 24 h before experimental procedures from consuming coffee, tea, alcohol or tobacco and doing vigorous exercise. Using simple random sampling, the participants were distributed in two groups, an experimental exercise group (EG: *n* = 19, 65.3 ± 4.7 years, 1.52 ± 4.12 m) that performed an METP during nine months, followed by twelve months of DT, and a control group (CG: *n* = 14, 66.4 ± 5.2 years, 1.54 ± 5.58 m) that did not perform the METP (Table 1).

#### 2.2.1. Exercise Program and Detraining Period

The nine-month exercise program consisted of an METP performed twice a week for about sixty minutes per session, with strength, flexibility, aerobic endurance and balance exercises prescribed by a sports sciences specialist [10,11,20]. The program sessions followed the same structure [18]: warm-up (5–8 min), aerobic exercise (15–25 min), resistance exercise (15–20 min) and cool down (5–10 min). The warm-up included low-intensity walking and mobility exercises followed by the aerobic exercises that included aerobic choreography. The resistance exercises were prescribed in a full-body circuit for strength, agility, mobility and coordination with a 20 to 30 s rest between sets (e.g.,: arm raise; air squat; heel to toe walking; back leg raises; obstacle overpass), together with social interaction among the elders. The sessions ended with a cool down, full-body stretching and relaxation.

METP intensity used was progressive; for the aerobic exercise, in the first month, we maintained it at 2–3, and in the next months we increased it up to 4–5 according to Rating of Perceived Exertion adapted scale [26], and for the resistance exercises, the first month was used for familiarization with and adaptation to the exercises and their techniques, and the next months we increased the series and repetitions from 2 to 4 and from 16 to 30, respectively.

After METP, the participants followed twelve consecutive months of detraining and were asked to maintain their normal lifestyles (dietary and physical activity routines) and to avoid regular exercise. During DT, they were monitored frequently and systematically by telephone.

#### 2.2.2. Body Composition and Hemodynamic and Lipid Profiles

Body mass (kg), body fat percentage (%BF, %) and body mass index (BMI, kg.m^−2^) were assessed through a bioelectrical impedance analysis (OMRON BF 303, Matsusaka, Japan), and height was assessed with a stadiometer (Seca, Hamburg, Germany) for the body compositions.

For the hemodynamic profiles, blood pressure and resting heart rate (RHR) were measured three times in a seated rest position with the left arm in support (AHA, 2005), with an Omron Digital Blood Pressure Monitor HEM-907 (Matsusaka, Japan).

The lipid profiles were collected according to the Diabetes Atlas Committee’s procedures for measurement of triglycerides (TG, mg/dL), total cholesterol (TC, mg/dL) and blood glucose (GL, mg/dL) with the use of Roche Diagnostics GmbH Cobas Accutrend Plus (Mannheim, Germany).

#### 2.2.3. Functional Capacity

The Rickli and Jones Senior Fitness Test [27] with six motor tests—lower (LBS; 30 s chair stand) and upper body strength (UBS; arm curl), lower (LBF; chair sit-and-reach) and upper body flexibility (UBF; back scratch), agility (2TUG; 8-foot up-and-go) and cardiorespiratory capacity (6MWT; 6-min walk test)—was used to assess participants’ FCs.

### 2.3. Statistical Analysis

Shapiro-Wilk test was used to check the normality of the data and to compare within and between groups all variables over time; we used a separate two-way ANOVA mixed model. Post hoc Bonferroni test was performed when necessary. For comparisons between groups, we used an independent student’s *t*-test with delta percentage of each variable. For delta percentage (∆%), we used the standard formula: ∆% = [(post-test score − pretest score)/pretest score] × 100. The significance level was set at *p* ≤ 0.05. All data analyses were performed with SPSS statistical software v. 20 (IBM Corp., Armonk, NY, USA). The sample-size calculation was performed using the program G*Power 3.1 with a power of 0.8; a total of 24 participants was required.

## 3. Results

For this study, the attendance rate for training sessions was 84%. Nine months of the METP resulted in improvements in body compositions and hemodynamic and lipid profiles (Table 1 and Table 2).

In relation to the hemodynamic profiles, changes over time demonstrated that SBP worsened their values after 12 months of detraining. DBP preserved their benefits over time through the full detraining period. RHR maintained training benefits for only a 3-month detraining period (Figure 1). For comparisons between groups, delta scores showed that SBP maintained the negative effect after 12 months of detraining, with higher values than the control condition. Conversely, DBP and RHR preserved its benefits in the group comparison (Table 2).

For lipid profiles, improvements in TG and TC lasted through 12 months of detraining. However, three months of detraining was enough to make the GLs regress to baseline values (Figure 1). Delta scores showed that, when compared with CG, EG decreased TG, TC and GL after 9 months of the exercise program, but increased significantly after 3 and 12 months of detraining (Table 2).

For functional capacity, the improvements observed after the METP lasted for 12 months of detraining (Figure 2). For comparisons between groups, delta scores showed that 9 months of the exercise program was effective in improving functional capacity. However, after 3 and 12 months of detraining, there was a decrease in this effect.

## 4. Discussion

The present twenty-one-month study demonstrates that a detraining of three to twelve months promotes negative effects on the blood pressures, triglycerides, total cholesterols and functional capacities of older women. Furthermore, it shows that some of the benefits of exercise are maintained after three and twelve months. This study complements the published work [18] with the objective of mapping the volunteers who drop out of research.

The METP resulted in the improvement of the study of health profiles of older women, similar to a study by Douda et al. [28] that reported increases in muscular strength, cardiorespiratory capacity and flexibility after periods of nine months of METP over five years and in the same direction as Monteagudo et al. [29] and Bezerra et al. [30]. Sobrinho et al. [20] showed only 14 weeks of an METP combined with flexibility training is enough to promote increases in the flexibility of inactive older women and improvements in blood pressure. The hemodynamic and lipid profiles improved after the METP and were in line with the results of studies (Table 3) [17,18,31,32,33,34,35,36,37] that concluded that exercise helps older women regulate their blood pressure, triglycerides and total cholesterol, such as a study by Tofas et al. [38] that reported significant improvements in SBP and DBP after 8 months of resistance training.

The cessation of exercise promoted negative effects in older women, showing that the decline in effects started to occur after three months and were prolonged over the twelve months. Modaberi et al. [39], stated that detraining periods of more than 8 weeks started to affect older participants, even in the more successful exercise programs. In Zhang et al.’s [40] study, they reported that four weeks of detraining after 8 weeks of brisk walking were enough to decline the benefits, but Vogler et al. [41] reported that twelve weeks of detraining after twelve weeks of resistance exercise maintained the improvements achieved. According to He et al. [42], the loss of the benefits from training in older adults is partially related to genes; the genetic variants in inter-individual variations influence the muscular losses after detraining periods, as does training intensity.

Coetssee and Terblanche [43] and Correa et al. [44], with more than twelve weeks of detraining after resistance training, reported that the benefits in muscle strength were maintained, similar to our results. In our study, we see that twelve months of detraining doubles the negative impact of three months of detraining on muscle strength, cardiorespiratory capacity and flexibility but not on flexibility that stabilizes after three months. Essain et al. [35] stated that after three months of detraining, strength and cardiorespiratory capacity is maintained compared with baseline values. The same occurred in a study by Martinez-Aldao et al. [15] with five months of detraining that reported no changes in the functional capacities of older women, with the exception of muscle strength that could be due to a higher value in strength that could induce a higher decline with detraining. Bezerra et al. [30] reported that after one year of the cessation of exercise, older women maintained their FCs, localized muscular endurance and handgrip strengths, and recommended that low-intensity activities should be performed to reduce the age-related decline in physical capacity. A study by Lee et al. [36] with the same 12-month detraining period reported that was not enough to reverse functional capacities to baseline values with the exception of upper limb flexibility, a result that is partially similar to our results.

The hemodynamic and lipid profiles declined after the first three months of detraining and maintained their values over the next months. Although TG, TC and GL maintained some of the benefits of METP, SBP and DPB returned to baseline values. With four weeks of detraining, Nolan et al. [45] found no changes in TG, TC and blood pressure, and with a longer period, i.e., three months of DT after nine months of METP, Esain et al. [46] reported no significant changes in TG and TC. One of the reasons for the results of these two studies may be the normal values at the baseline and after the exercise program of the participants compared to ours that had higher values of TG and TC at the baseline. Moker et al. [47], with only two weeks of detraining after 6 months of exercise in middle-aged prehypertensive participants with mild to moderate dyslipidemia, reported that the benefits that occurred with exercise disappeared. Nascimento et al. [37], with 14 weeks of detraining after resistance training, reported that blood pressure improved with training, and benefits were maintained after the detraining period. Comparing these results, we can suggest that older women with pre- to established hypertension can have different reactions to exercise cessation compared with normotensive older women [18].

The limitations of our study resulted from dietary routine, which was not controlled during the entire study, and even though we contacted all the participants regularly during the detraining period, we did not quantify their physical activity levels.

## 5. Conclusions

The METP resulted in the improvement of the health profiles of hypertensive older women, but one year of exercise cessation compromised these benefits. Indeed, the first three months of detraining resulted in substantial negative changes in almost all the health profiles analyzed and was responsible for the major negative impact, although one year of detraining resulted in larger cumulative negative effects for the health of these hypertensive older women with high levels of lipidic variables.

## Figures and Tables

**Figure 1 ijerph-19-02871-f001:**
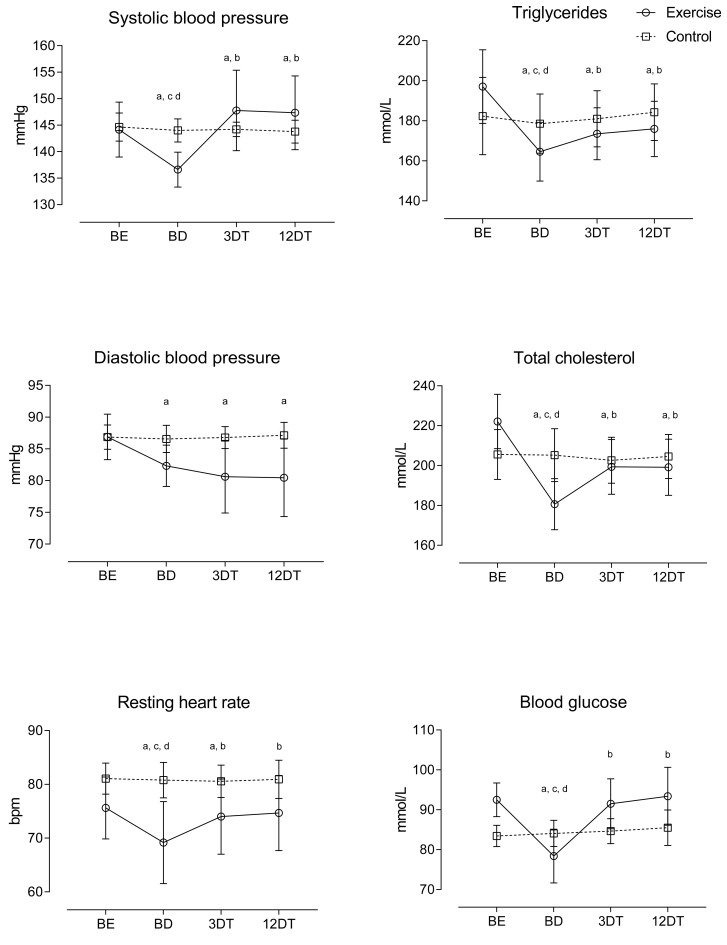
Relative changes in hemodynamic and lipidic profiles during the METP and the DT period; baseline (BE), after METP (BD), after three months of DT (3DT) and after 12 months of DT (12DT); millimoles per liter (mmol/L); ^a^
*p* < 0.05, significant difference between BEs for METP; ^b^
*p* < 0.05, significant difference between BDs for METP; ^c^
*p* < 0.05, significant difference between 3DTs for METP; ^d^
*p* < 0.05, significant difference between 12DTs for METP.

**Figure 2 ijerph-19-02871-f002:**
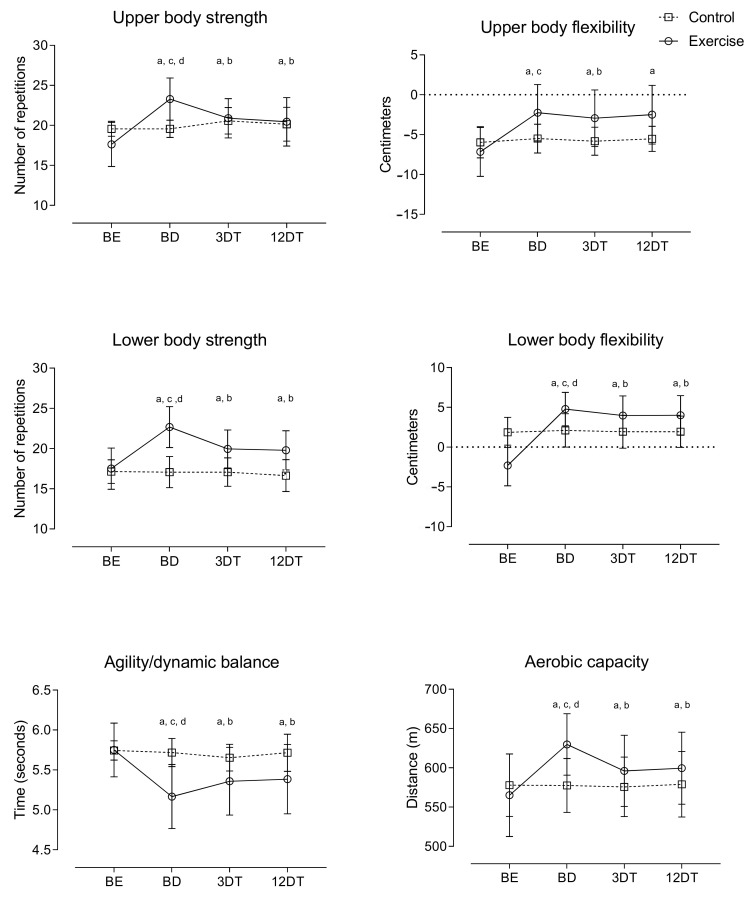
Relative changes in functional capacity during the METP and the DT period; baseline (BE), after METP (BD), after three months of DT (3DT) and after 12 months of DT (12DT); ^a^
*p* < 0.05, significant difference between BEs for METP; ^b^
*p* < 0.05, significant difference between BDs for METP; ^c^
*p* < 0.05, significant difference between 3DTs for METP; ^d^
*p* < 0.05, significant difference between 12DTs for METP.

**Table 1 ijerph-19-02871-t001:** Body compositions after 9 months of exercise program and over 12 months of detraining (mean ± SD).

Variable	CG (*n* = 14)				EG (*n* = 18)			
	BE	BD	3DT	12DT	BE	BD	3DT	12DT
Height (cm)	156.21 ± 3.10	156.21 ± 3.10	156.21 ± 3.10	156.21 ± 3.10	153.03 ± 3.42	153.03 ± 3.42	153.03 ± 3.42	153.03 ± 3.42
Body weight (kg)	69.38 ± 5.24	69.21 ± 5.55	69.79 ± 5.55 ^b^	69.86 ± 5.84	73.24 ± 7.30	71.84 ± 7.45 ^a^	72.24 ± 7.41 ^a,b^	72.28 ± 7.33 ^a^
%BF (%)	38.71 ± 1.12	39.05 ± 1.43	39.09 ± 1.14 ^a^	39.06 ± 1.34	39.00 ± 1.36	38.06 ± 1.28 ^a^	38.34 ± 1.33 ^a^	38.50 ± 1.34 ^a^
BMI (kg/m^2^)	28.46 ± 2.34	28.39 ± 2.50	28.63 ± 2.49 ^b^	28.65 ± 2.56	31.28 ± 3.04	30.68 ± 3.10 ^a^	30.85 ± 3.08 ^a,b^	30.87 ± 3.05 ^a^

BE: before exercise; BD: beginning of DT; 3DT: after three months of DT; 12DT: after twelve months of DT; ^a^ *p* < 0.05 compared with BE within group; ^b^
*p* < 0.05 compared with BD within group.

**Table 2 ijerph-19-02871-t002:** METP and DT effects (∆% mean ± SD).

	(∆ METP)			(∆ 3DT)			(∆ 12DT)		
Variable	CG	EG	*p Value*	CG	EG	*p Value*	CG	EG	*p Value*
Body weight (%)	−0.27 ± 1.44	−1.95 ± 0.76	<0.01	0.84 ± 0.52	0.57 ± 0.22	0.09	0.91 ± 1.19	0.64 ± 1.27	0.82
BMI (%)	−0.27 ± 1.44	−1.95 ± 0.76	<0.01	0.84 ± 0.52	0.57 ± 0.22	0.09	0.90 ± 1.18	0.64 ± 1.27	0.88
BF% (%)	0.88 ± 2.13	−2.42 ± 0.49	<0.01	0.16 ± 2.42	0.76 ± 0.80	0.39	0.05 ± 2.29	1.18 ± 1.65	0.14
SBP (%)	−0.42 ± 1.85	−5.17 ± 2.70	<0.01	0.17 ± 1.91	8.19 ± 5.17	<0.01	−0.12 ± 2.46	7.86 ± 4.62	<0.01
DBP (%)	−0.32 ± 1.78	−5.23 ± 1.36	<0.01	0.26 ± 1.13	−2.10 ± 6.13	0.13	0.67 ± 1.27	−2.27 ± 6.74	0.09
RHR (%)	−0.35 ± 1.90	−8.72 ± 3.96	<0.01	−0.25 ± 1.29	7.21 ± 3.30	<0.01	0.18 ± 1.76	8.19 ± 3.86	<0.01
TG (%)	−1.81 ± 4.29	−16.41 ± 3.39	<0.01	1.46 ± 3.00	5.66 ± 4.46	<0.01	3.45 ± 6.48	7.25 ± 7.21	0.08
TC (%)	−0.19 ± 1.37	−18.71 ± 1.99	<0.01	−1.16 ± 3.16	10.42 ± 3.09	<0.01	−0.17 ± 4.43	10.38 ± 4.72	<0.01
GL (%)	0.77 ± 2.34	−15.25 ± 6.34	<0.01	0.73 ± 3.12	17.12 ± 7.59	<0.01	1.72 ± 4.12	19.36 ± 6.01	<0.01
LBS (%)	−0.58 ± 5.11	30.35 ± 8.67	<0.01	0.46 ± 8.55	−12.03 ± 2.99	<0.01	−2.39 ± 6.01	−12.75 ± 4.73	<0.01
UBS (%)	0.00 ± 2.86	33.62 ± 12.87	<0.01	5.43 ± 10.57	−9.86 ± 8.84	<0.01	3.16 ± 11.67	−11.68 ± 12.55	<0.01
2TUG (%)	−0.45 ± 2.18	−10.13 ± 5.01	<0.01	−1.10 ± 2.12	3.78 ± 3.87	<0.01	−0.02 ± 3.70	4.37 ± 5.67	0.02
6MWT(%)	0.02 ± 2.22	11.93 ± 6.83	<0.01	−0.34 ± 1.48	−5.38 ± 3.49	<0.01	0.21 ± 2.79	−4.82 ± 4.38	<0.01
UBF (cm)	0.46 ± 0.63	4.92 ± 1.65	<0.01	−0.32 ± 0.58	−0.69 ± 0.35	0.03	−0.04 ± 1.42	−0.25 ± 1.23	0.65
LBF (cm)	0.25 ± 1.28	7.08 ± 2.43	<0.01	−0.18 ± 0.54	−0.81 ± 1.20	0.05	−0.17 ± 0.69	−0.77 ± 1.26	0.10

EG (*n* = 18); CG (*n* = 14); BMI: body mass index; BF: body fat; SBP: systolic blood pressure; DBP: diastolic blood pressure; RHR: resting heart rate; TG: triglycerides; TC: total cholesterol; GL: blood glucose; LBS: lower body strength; UBS: upper body strength; 2TUG: agility/dynamic balance; 6MWT: aerobic endurance six-minute walk test; UBF: upper body flexibility; LBF: lower body flexibility.

**Table 3 ijerph-19-02871-t003:** METP and detraining effects in older adults in other studies.

**Study**	**N**	**Type of Training**	**Intensity and Duration of Training**	**Effects of Training**	**Detraining Duration**	**Effects of Detraining**
Bezerra et al. [30].	15	Strength training	3 sessions/week 9 weeks	Functional capacity and strength improved	1 year	Strength and functional capacity benefits maintained
Douda et al. [28]	42	METP	3 × 45 min/week 9 months every year over 5 years	Functional capacity and strength improved	3 months every year over 5 years	Functional capacity and strength benefits maintained
Essain et al. [35]	38	METP	2 × 50 min/week 9 months	-	3 months	TG, TC, strength and cardiorespiratory benefits maintained
Lee et al. [36]	18	METP	2 s 60 min/week 12 months	Functional capacity improved	12 months	Functional capacity benefits maintained except with upper-body flexibility
Leitão et al. [18]	17	METP	2 × 45 min/week 9 months	Functional capacity, TG and TC improved	3 months	Functional capacity, TG and TC benefits maintained
Martinez-Aldao et al. [15]	65	METP	2 × 45 min/week 8 months	-	5 months	Strength, upper body flexibility and agility decreased
Nascimento et al. [37]	12	Strength training	2 sessions/week	Strength and BP improved	14 weeks	Strength and BP benefits maintained
Sobrinho et al. [20]	52	METP	2 × 90 min/week 14 weeks	Flexibility and BP improved	-	-

## Data Availability

Available through the corresponding author by request.

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
