# Peer review of "Three-Month vs. One-Year Detraining Effects after Multicomponent Exercise Program in Hypertensive Older Women"

_ijerph, 2022, doi:10.3390/ijerph19052871_

Round 1

Reviewer 1 Report

Firstly, I would like to congratulate the authors for the work/job and topic addressed in this manuscript. The effort involved in carrying out this type of intervention in humans with such a long duration must be recognized. Secondly, I think it is very important to make visible the problems that a sedentary lifestyle entails as well as stopping training programs.

Below, I am going to make a couple of suggestions that I hope will improve your work.

Abstract

Lines 30 and 32: before writing the abbreviations referring to a term, the full name of the term must be included. Example: Experimental group (EG).

Introduction

In the paragraph discussing the effects of exercise, more information about other types of exercise such as strength work or interval training should be included. I also think that the intensity of work recommended for this population should be alluded, since this is an important factor to take into account.

Methodology

  • Indicate how the distribution of the participants to each group was carried out. Was it a blinding?
  • Indicate how progress was made and how the load and intensity of the exercises was controlled.
  • Line 133. The first time a term appears in the text, it must be written completely.
  • After reviewing one on their manuscript previously published in this same journal “Multicomponent Exercise Program for Improvement of Functional Capacity and Lipidic Profile of Older Women with High Cholesterol and High Triglycerides”, a great similarity in the methodology section in both papers has been observed.  I consider that both manuscripts are related. However, from my point of view, the authors should modify the wording to eliminate the similarity. 

Results

You should clarify the instructions given to the participants during the detraining period. Were they prohibited from doing any type of physical activity or participating in training sessions? At the end of the manuscript, you expose the lack of control of the physical activity level during this period as a limitation of the study. However, you must clarify exactly what instructions were given to the participants, since prohibiting the practice of physical activity for such a long period might have very negative consequences for this population.

Due to there is a comparison between the control and experimental groups, did you consider whether there were differences between the initial values? If so, it should be included and explained. If you have not taken this factor into account, you should explain in the work that the results presented in Table 2 may be conditioned by the initial state of the participants.

Author Response

We are grateful for your consideration of this manuscript, and we also very much appreciate your suggestions, which have been very helpful in improving the manuscript. We also thank the reviewers for their careful reading of our text. All the comments we received on this study of all reviewers have been attended into account in improving the quality of the article, and we present our reply to each of them separately.

Abstract

Lines 30 and 32: before writing the abbreviations referring to a term, the full name of the term must be included. Example: Experimental group (EG).

A: We changed

Introduction

In the paragraph discussing the effects of exercise, more information about other types of exercise such as strength work or interval training should be included. I also think that the intensity of work recommended for this population should be alluded, since this is an important factor to take into account.

A: We add a paragraph. Line 59-61.

Methodology

  • Indicate how the distribution of the participants to each group was carried out. Was it a blinding?

A: We add sentence Line 109. We used a simple random sampling.

  • Indicate how progress was made and how the load and intensity of the exercises was controlled.

A: We had sentence. LINE 125-129

  • Line 133. The first time a term appears in the text, it must be written completely.

A: We changed

  • After reviewing one on their manuscript previously published in this same journal “Multicomponent Exercise Program for Improvement of Functional Capacity and Lipidic Profile of Older Women with High Cholesterol and High Triglycerides”, a great similarity in the methodology section in both papers has been observed.  I consider that both manuscripts are related. However, from my point of view, the authors should modify the wording to eliminate the similarity. 

A: We rewrite the methods to eliminate the similarity.

Results

You should clarify the instructions given to the participants during the detraining period. Were they prohibited from doing any type of physical activity or participating in training sessions? At the end of the manuscript, you expose the lack of control of the physical activity level during this period as a limitation of the study. However, you must clarify exactly what instructions were given to the participants, since prohibiting the practice of physical activity for such a long period might have very negative consequences for this population.

A: We agree with you. We did not prohibited the older women to exercise, we just tell to avoid (LINE130-133) and according to our contact they did it. We did not controlled the physical activity because we did not have enough accelerometers.

Due to there is a comparison between the control and experimental groups, did you consider whether there were differences between the initial values? If so, it should be included and explained. If you have not taken this factor into account, you should explain in the work that the results presented in Table 2 may be conditioned by the initial state of the participants.

A: We have described the delta-value comparisons since our idea was to show the changes in physiological parameters due to the training period instead of only showing absolute values.

Reviewer 2 Report

General comments:

The introduction of the manuscript is generally well-written. The authors reported relevant and up to date literature. However, some important information is missing. My main concerns are listed below.

Specific comments

  • Please specify your inclusion criteria. What was your blood pressure threshold? Did you exclude participants with very high blood pressure levels? Did you measure blood pressure ones or twice, in a sitting or supine position or in 24h setting? More information is needed to interpret your inclusion criteria and therefore your study population.
  • What is meant by “medical contraindication”. Please specify.
  • Please specify your randomization procedure.
  • Figure 2 would highly benefit from clear headlines for every figure instead of abbreviations to increase readability.
  • No information about drop-outs are presented. Please include a flow-chart to present drop-outs and their reasons at every timepoint of your study.
  • Did the authors calculate the power of this study by analyzing the sample size needed for the main outcome? Was this sample size reached?

Minor concerns

  • I would recommend to report body mass instead of height to describe your groups in the abstract and first part of the results.
  • Try to avoid very long sentences like lines 51-55.

Author Response

We are grateful for your consideration of this manuscript, and we also very much appreciate your suggestions, which have been very helpful in improving the manuscript. We also thank the reviewers for their careful reading of our text. All the comments we received on this study of all reviewers have been attended into account in improving the quality of the article, and we present our reply to each of them separately.

Specific comments

  • Please specify your inclusion criteria. What was your blood pressure threshold? Did you exclude participants with very high blood pressure levels? Did you measure blood pressure ones or twice, in a sitting or supine position or in 24h setting? More information is needed to interpret your inclusion criteria and therefore your study population.

A: We add in the inclusion criteria the values of blood pressure threshold. There were no very high blood pressure participants in our sample. The procedures for blood pressure assessment are described in line 141-143.

  • What is meant by “medical contraindication”. Please specify.

A: We meant by medical contraindication the negative opinion by the doctor to the patient to perform exercise (e.g., surgery). LINE 99

  • Please specify your randomization procedure.

A: We add sentence Line 109. We used a simple random sampling.

  • Figure 2 would highly benefit from clear headlines for every figure instead of abbreviations to increase readability.

A: We changed

  • No information about drop-outs are presented. Please include a flow-chart to present drop-outs and their reasons at every timepoint of your study.

A: We add sentence. There were 7 drop-outs due to low adherence to the sessions (less than 75%) LINE 105-106

  • Did the authors calculate the power of this study by analyzing the sample size needed for the main outcome? Was this sample size reached?

A: Thank you for your question. Using an a priori sample-size calculation (with a power of 0.8), a total of 24 participants were required. However, we included more fifteen subjects. Nevertheless, there were 7 drop-outs. Therefore, thirty-two functionally independent older women voluntarily participated in this research. We have described the sample-size calculation in the materials and methods section.

Minor concerns

  • I would recommend to report body mass instead of height to describe your groups in the abstract and first part of the results.

A: We changed

  • Try to avoid very long sentences like lines 51-55.

A: We rewrite the sentence.

Round 2

Reviewer 1 Report

The authors have made an effort to improve the manuscript and make all suggested changes.
My decision is to accept the manuscript

Author Response

Dear reviewer,

we are grateful for your consideration of this manuscript, and we also very much appreciate your suggestions, which have been very helpful in improving the manuscript.

best regards

Reviewer 2 Report

  • The authors submitted a study with hypertensive participants. So hypertension is an inclusion criteria and have to be exactly described. The authors wrote: "The participants inclusion criteria were values above normal blood pressure levels (SBP: <130 mm Hg; DBP: <80 mm Hg), can practice exercise without contraindication and attendance to at least 75% of the sessions." --> These blood pressure thresholds are based on 24h BP measurements defined by the ESC. The authors did not measure BP during 24h or? I would recommend writing: "The participants' inclusion criteria were values above normal blood pressure levels (study threshold: SBP: >=130 mmHg; ??and/or?? DBP: >=80 mmHg), can practice exercise without contraindication and attendance to at least 75% of the sessions." Please specify this important part of your inclusion criteria.
  • The authors answered based on my comment A: Thank you for your question. Using an a priori sample-size calculation (with a power of 0.8), a total of 24 participants were required. However, we included more fifteen subjects. Nevertheless, there were 7 drop-outs. Therefore, thirty-two functionally independent older women voluntarily participated in this research. We have described the sample-size calculation in the materials and methods section. --> I am sorry but I can not find your description of the sample-size calculation in your manuscript.

Author Response

Dear Reviewer,

we are grateful for your consideration of this manuscript, and we also very much appreciate your suggestions, which have been very helpful in improving the manuscript.

We add your last suggestions to the article.

  • The authors submitted a study with hypertensive participants. So hypertension is an inclusion criteria and have to be exactly described. The authors wrote: "The participants inclusion criteria were values above normal blood pressure levels (SBP: <130 mm Hg; DBP: <80 mm Hg), can practice exercise without contraindication and attendance to at least 75% of the sessions." --> These blood pressure thresholds are based on 24h BP measurements defined by the ESC. The authors did not measure BP during 24h or? I would recommend writing: "The participants' inclusion criteria were values above normal blood pressure levels (study threshold: SBP: >=130 mmHg; ??and/or?? DBP: >=80 mmHg), can practice exercise without contraindication and attendance to at least 75% of the sessions." Please specify this important part of your inclusion criteria.

A: We add "(study threshold: SBP:≥130 mmhg; and DBP≥80 mmHg)". Line 96

  • The authors answered based on my comment A: Thank you for your question. Using an a priori sample-size calculation (with a power of 0.8), a total of 24 participants were required. However, we included more fifteen subjects. Nevertheless, there were 7 drop-outs. Therefore, thirty-two functionally independent older women voluntarily participated in this research. We have described the sample-size calculation in the materials and methods section. --> I am sorry but I can not find your description of the sample-size calculation in your manuscript.

A: We add "The sample-size calculation was performed using the G*Power 3.1 program with a power of 0.8, a total of 24 participants were required." Line 163-165